# MASKED PRETRAINING FOR MULTI-AGENT DECISION MAKING

## ABSTRACT

Building a single generalist agent with zero-shot capability has recently sparked significant advancements in decision-making. However, extending this capability to multi-agent scenarios presents challenges. Most current works struggle with zero-shot capabilities, due to two challenges particular to the multi-agent settings: a mismatch between centralized pretraining and decentralized execution, and varying agent numbers and action spaces, making it difficult to create generalizable representations across diverse downstream tasks. To overcome these challenges, we propose a **Mask**ed pretraining framework for **M**ulti-**a**gent decision making (MaskMA). This model, based on transformer architecture, employs a mask-based collaborative learning strategy suited for decentralized execution with partial observation. Moreover, MaskMA integrates a generalizable action representation by dividing the action space into actions toward self-information and actions related to other entities. This flexibility allows MaskMA to tackle tasks with varying agent numbers and thus different action spaces. Extensive experiments in SMAC reveal MaskMA, with a single model pretrained on 11 training maps, can achieve an impressive 77.8% zero-shot win rate on 60 unseen test maps by decentralized execution, while also performing effectively on other types of downstream tasks (*e.g.,* varied policies collaboration and ad hoc team play).

## 1 INTRODUCTION

Foundation model is a large model trained on vast data and can easily generalize across various downstreaming tasks in natural language processing, called emergent behavior. The powerful foundation models (Ouyang et al., 2022; Touvron et al., 2023; Brown et al., 2020; Ramesh et al., 2022; Rombach et al., 2022; Radford et al., 2021) bring artificial intelligence techniques to the daily life of people, serving as the assistant to boost the development of various industries. The reinforcement learning community (Chen et al., 2021; Carroll et al.; Liu et al.; Janner et al., 2021; 2022) has shown a growing interest in designing simple yet effective foundation models and training strategies tailored to decision-making. A natural follow-up question is how to build a foundation model that serves as a single generalist agent with strong zero-shot capability for multi-agent decision-making.

Compared to single-agent scenarios, directly utilizing transformers for centralized pretraining in multi-agent settings encounters two primary challenges. (1) Mismatch between centralized pretraining and decentralized execution. Multi-agent decision-making typically follows centralized training with a decentralized execution approach. However, transformers, as a centralized training architecture, utilize all units as inputs. This misaligns with the decentralized execution phase where each agent's perception is limited to only nearby units, significantly impacting performance. (2) Varying numbers of agents and actions. Downstream tasks have different numbers of agents, resulting in varying action spaces. Most existing methods treat multi-agent decision-making as a sequence modeling problem and directly employ transformer architectures, often overlooking or inadequately addressing the aforementioned challenges. For instance, MADT (Meng et al., 2021) circumvents the mismatch challenge by transforming multi-agent pretraining data into single-agent pretraining data and adopting decentralized pretraining with decentralized execution, but this comes at the expense of not fully utilizing the information from all agents during the pretraining stage. Regarding the issue of different action spaces caused by varying agent numbers, MADT takes a simplistic approach by setting a large action space and muting the unavailable actions using an action mask. However, this method

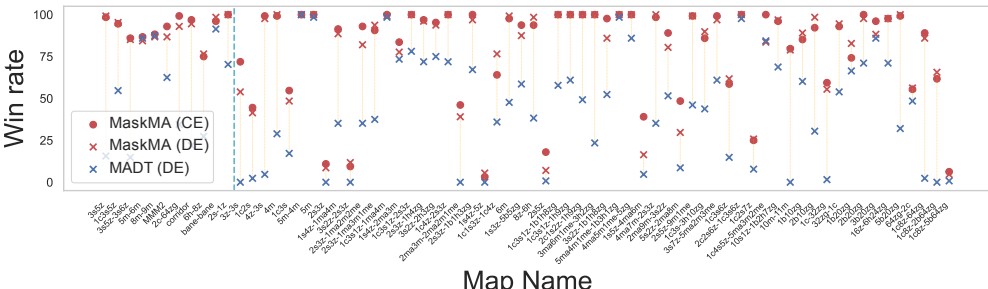

Figure 1: **Win rate on training and test maps.** The dashed line (blue) separates the 11 training maps on the left from the 60 test maps on the right. The orange line represents the performance difference between MaskMA and MADT, showcasing how MaskMA outperforms MADT by up to 92.97%.

suffers from poor generalization because the same component of the action vector represents different physical meanings in tasks with different numbers of agents.

In response, we propose two scalable techniques: a Mask-based Collaborative Learning Strategy (MCLS) and a Generalizable Action Representation (GAR). The two techniques form the basis of a new masked pretraining framework for multi-agent decision-making, named MaskMA. To address the first challenge, we present a transformer with MCLS by incorporating random masking into the attention matrix of the transformer, effectively reconciling the discrepancy between centralized pretraining and partial observations and bolstering the model's generalization capabilities. To handle the second challenge, MaskMA integrates GAR by categorizing actions into those directed toward the environment and those involving interactions with other units. The former relies solely on self-information, and the latter depends on their interrelationships, respectively. This approach allows MaskMA to excel across tasks with varying agent numbers and action spaces.

We evaluate MaskMA's performance using the StarCraft Multi-Agent Challenge (SMAC) benchmark. To validate the potential of zero-shot, we provide a challenging setting, using only 11 maps for training and 60 maps for testing. Extensive experiments demonstrate that our model significantly outperforms the previous state-of-the-art in zero-shot scenarios. We also provide various downstream tasks to further evaluate the strong generalization of MaskMA, including varied policies collaboration, teammate malfunction, and ad hoc team play. This work lays the groundwork for further advancements in multi-agent fundamental models, with potential applications across a wide range of domains.

Our main contributions are as follows:

1. We introduce the masked pretraining framework for multi-agent decision-making (MaskMA), which pre-trains a transformer architecture with a mask-based collaborative learning strategy (MCLS) and a generalizable action representation (GAR).

2. To test MaskMA's performance, we set up 1) a challenging zero-shot task: training on only 11 maps and testing on 60 different maps in the SMAC (Samvelyan et al., 2019), and 2) three downstream tasks including varied policies collaboration, teammate malfunction, and ad hoc team play.

3. MaskMA is the **first** multi-agent pretraining model for decision-making with strong zero-shot performance. MaskMA, using a **single** model pre-trained on 11 training maps, achieves an impressive 77.8% zero-shot win rate on 60 unseen test maps by decentralized execution.

## 2 RELATED WORK

**Decision Making as Sequence Modeling Problem and Pretraining** In recent years, the integration of sequence modeling into decision-making paradigms has emerged as a promising avenue for enhancing reinforcement learning strategies. DT (Chen et al., 2021) casts the reinforcement learning as a sequence modeling problem conditioned on return-to-go, using a transformer to generate optimal action. MaskDP (Liu et al.) utilizes autoencoders on state-action trajectories, learning the environment's dynamics by masking and reconstructing states and actions. Uni[MASK] (Carroll et al.) expresses various tasks as distinct masking schemes in sequence modeling, using a single model trained with randomly sampled maskings. In this paper, we explore the design of sequences in MARL and how it can be made compatible with the mask-based collaborative learning strategy.

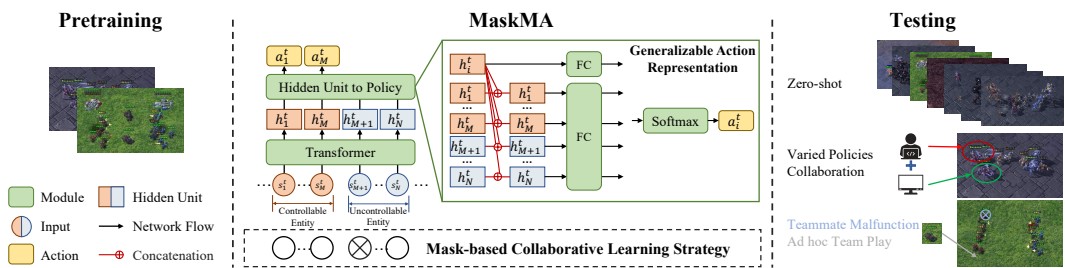

Figure 2: **MaskMA.** MaskMA employs the transformer architecture combined with generalizable action representation trained using a mask-based collaborative learning strategy. It effectively generalizes skills and knowledge from training maps into various downstream tasks, including unseen maps, varied policies collaboration, teammate malfunction, and ad hoc team play.

**MARL as Sequence Modeling Problem**    Recently several works collectively contribute to the understanding of MARL as a sequence modeling problem. MADT (Meng et al., 2021) introduces Decision Transformer (Chen et al., 2021) into MARL, significantly improving sample efficiency and achieving strong performance in both few-shot and zero-shot cases in SMAC. MAT (Wen et al., 2022) leverages an encoder-decoder architecture, incorporating the multi-agent advantage decomposition theorem to reduce the joint policy search problem into a sequential decision-making process. Tseng et al. (2022) utilize the Transformer architecture and propose a method that identifies and recombines optimal behaviors through a teacher policy. ODIS (Zhang et al., 2023) trains a state encoder and an action decoder to extract task-invariant coordination skills from offline multi-task data. In contrast, our proposed MaskMA adapts the Transformer architecture to MARL by designing a sequence of inputs and outputs for a generalizable action representation. This approach offers broad generalizability across varying actions and various downstream tasks.

**Action Representation.**    Recent works have explored semantic action in multi-agent environments. ASN (Wang et al.) focuses on modeling the effects of actions by encoding the semantics of actions to understand the consequences of agent actions and improve coordination among agents. UPDeT (Hu et al., 2021) employs a policy decoupling mechanism that separates the learning of local policies for individual agents from the coordination among agents using transformers. In contrast, MaskMA emphasizes sequence modeling and masking strategies, focusing on the correlation between agents taking actions. While UPDeT concentrates on policy decoupling for improved coordination among agents and ASN is centered on modeling the effects of actions and their interactions in multi-agent environments, MaskMA aims to learn more generalizable skills from training maps, which can be applied to a wide range of downstream tasks. This unique approach allows MaskMA to excel in scenarios involving varied policies collaboration, teammate malfunction, and ad hoc team play.

## 3  METHOD

To achieve zero-shot generalization in multi-agent decision-making tasks, where the agents need to cooperate and learn effective strategies to adapt to various scenarios, we propose MaskMA, a masked pretraining framework for multi-agent decision-making, by leveraging the transformer with generalizable action representation to capture the underlying correlations among agents and their actions while maintaining adaptability to dynamic scenarios.

Agents are subject to partial observation in multi-agent tasks, *i.e.*, each agent has limited sight and can only observe part of other agents and other units (e.g., enemy to defeat) in the environment. Existing works, such as those proposed in (Liu et al.) and (Hu et al., 2021), typically train each agent's policy independently. Specifically, the input to each agent's policy is its own observation. Such an independent learning pipeline leads to an increase in computational complexity of $O\left(N^3\right)$ w.r.t agent numbers $N$. To address these challenges, we introduce Mask-based Collaborative Learning, which employs random masking to train the policies collaboratively, aligning well with the partial observation.

Table 1: **Win rate on training maps.** The offline datasets consist of 10k or 50k expert trajectories per map collected by specific expert policies. With the mask-based collaborative learning strategy, MaskMA consistently demonstrates high performance in both centralized execution (CE) and decentralized execution (DE) settings. Furthermore, MaskMA's generalizable action representation allows it to easily adapt and converge on maps with diverse characteristics. In contrast, MADT struggles to handle different action spaces and achieves a win rate of only 51.78% even after extensive training.

| Map_name | # Episodes | Return Distribution | Ours | | MADT |
| --- | --- | --- | --- | --- | --- |
| | | | CE | DE | DE |
| 3s_vs_5z | 50k | 19.40±1.89 | 85.94±3.49 | 82.81±7.81 | 73.44±3.49 |
| 3s5z | 10k | 18.83±2.48 | 98.44±1.56 | 99.22±1.35 | 15.62±6.99 |
| 1c3s5z | 10k | 19.51±1.40 | 94.53±4.06 | 95.31±1.56 | 54.69±8.41 |
| 3s5z_vs_3s6z | 10k | 19.69±1.27 | 85.94±6.44 | 85.16±5.58 | 14.84±9.97 |
| 5m_vs_6m | 10k | 18.37±3.69 | 86.72±1.35 | 84.38±4.94 | 85.94±5.18 |
| 8m_vs_9m | 10k | 19.12±2.57 | 88.28±6.00 | 86.72±4.06 | 87.50±2.21 |
| MMM2 | 50k | 18.68±3.42 | 92.97±2.59 | 86.72±4.62 | 62.50±11.69 |
| 2c_vs_64zg | 10k | 19.87±0.48 | 99.22±1.35 | 92.97±2.59 | 34.38±9.11 |
| corridor | 10k | 19.44±1.61 | 96.88±3.83 | 94.53±2.59 | 21.88±11.48 |
| 6h_vs_8z | 10k | 18.72±2.33 | 75.00±5.85 | 76.56±6.44 | 27.34±6.77 |
| bane_vs_bane | 10k | 19.61±1.26 | 96.09±2.59 | 98.44±1.56 | 91.41±4.62 |
| average | ∼ | 19.20±2.04 | 90.91±3.56 | 89.35±3.92 | 51.78 ±7.27 |

## 3.1 FORMULATION

We exploit a decentralized partially observable Markov decision process (Oliehoek & Amato, 2015) to define a cooperative multi-agent task, denoted as $G = <S, U, A, P, O, r, \gamma>$. Here $S$ represents the global state of the environment, and $U \triangleq \{u_1, u_2, ..., u_N\}$ denotes the set of $N$ units, where the first $M$ units are the agents controlled by the policy and the rest $N - M$ units are uncontrolled units in the environment. $A = A_1 \times A_2 \times ... \times A_M$ is the action space for controllable units. At time step $t$, each agent $u_i \in \{u_1, u_2, ..., u_M\}$ selects an action $\boldsymbol{a}_i \in A_i$, forming a joint action $\boldsymbol{a} \in A$. The joint action $\boldsymbol{a}$ at state $s \in S$ triggers a transition of $G$, subject to the transition function $P(s' \mid s, \boldsymbol{a}) : S \times A \times S \rightarrow [0, 1]$. Meanwhile, a shared reward function $r(s, \boldsymbol{a}) : S \times A \rightarrow \mathbb{R}$, with $\gamma \in [0, 1]$ denoting the discount factor. We consider a partially observable setting in which each agent $u_i$ makes individual observations $o_i$ to the observation function $o_i = O(s, u_i)$.

## 3.2 MASK-BASED COLLABORATIVE LEARNING

We utilize a standard causal transformer with only encoder layers as our model backbone. The input is the recent $L$ global states $s^{t-L+1}, s^{t-L+2}, ..., s^t$. We define $s^t = \{s^t(u_1), s^t(u_2), ..., s^t(u_N)\}$, *i.e.*, $s^t$ is the union of the states of all units at $t$-th time step. At the input, the state $s^t(u_i)$ of each unit $u_i$ at each time step $t$ corresponds to a token, resulting in total $L \times N$ tokens. Note that $s^t(u_i)$ only contains the state of the entity itself and does not include any information about other entities. For example, in SMAC, $s^t(u_i)$ includes unit type, position, health, shield, and so on.

We define the local observation $o_i^t$ of each unit $u_i$ as the states of all units observed by unit $u_i$ at $t$-th step, namely $o_i^t = \{s^t(u_i) \mid i \in p_i^t\}$, with $p_i^t$ denoting the indexes of units observable to $u_i$. Previous methods independently learn the policy of each unit $u_i$ with their corresponding $o_i^t$ as the input. On the contrary, in this paper, we propose to randomly mask part of the units in $s^t$ and collaboratively learn the policies of unmasked units. Formally, we randomly select part of the units in $s^t$ for each step $t$ of the $L$ input steps of states, represented by $\hat{s}^t = \{s^t(u_i) \mid i \in m^t\}$, and learns the policies of the units $u_i$ in $m^t$ with supervised learning.

Specifically, we utilize the attention matrix to implement mask-based collaborative learning. We define the original attention mask matrix $m_o$, the mask matrix $m_r$ with elements that have a certain probability of being 1, the final mask matrix $m$ used by MaskMA, as well as some intermediate matrices $m_1$, $m_2$, $R$ and $J_2$. The shape of these mask matrices is $(LN \times LN)$, corresponding to $L \times N$ input tokens. We will proceed with the following steps to obtain $m$.

Table 2: **Win rate on test maps.** We assessed the performance of MaskMA and other baseline models on a collection of 60 unseen test maps. These models were trained using a set of 11 training maps. The term "Entity" denotes the number of entities present in each individual map, while "Map Numbers" represents the number of maps that fulfill certain conditions. The results demonstrate that MaskMA is an excellent zero-shot learner.

| Entity | Map Numbers | Ours | | MADT |
|---|---|---|---|---|
| | | CE | DE | DE |
| $\leq 10$ | 23 | 76.26±3.30 | 74.38±3.57 | 43.55±3.94 |
| $10 \sim 20$ | 22 | 83.81±2.85 | 80.08±2.98 | 46.77±3.67 |
| $> 20$ | 15 | 79.01±5.02 | 79.48±3.84 | 39.53±3.61 |
| All | 60 | 79.71±3.56 | 77.75±3.42 | 43.72±3.76 |

For multi-agent sequential modeling, the mask is casual in the timestep dimension and non-casual within each timestep. Therefore, we have $m_1 = \text{Diag}(J_1, J_1, ..., J_1)$, where $J_1$ is an $N \times N$ matrix filled with ones, and $\text{Diag}$ constructs a diagonal matrix $m_1$. Then we get $m_2 = \text{Tri}(J_2)$, where $J_2$ is a matrix filled with ones, and $\text{Tri}$ represents the operation of extracting the lower triangular part. Finally, we get $m_o = m_1 \vee m_2$. Define the mask ratio as $r$, and generate the mask matrix $m_r = R >= r$, where $R$ is a matrix obtained by uniform sampling elements from 0 to 1. Then we get the final mask matrix $m = m_o \wedge m_r$. We explore different types of masks, including a set of fixed mask ratios, environment mask, and random mask ratios chosen from $(0, 1)$ for units at each time step. We observe that the implementation of the random mask strategy, which encompasses different fixed ratios and mask types, leads to the acquisition of meaningful skills and knowledge applicable to various downstream tasks.

**Execution** We can efficiently shift between centralized and decentralized execution by adjusting the attention mask matrix $m$. For decentralized execution we alter $m$ so that each agent only considers surrounding agents during self-attention, while for centralized execution we set $m$ as $m_o$.

### 3.3 GENERALIZABLE ACTION REPRESENTATION

We harnessed the transformer's capability to handle variable-length tokens, *i.e.*, the architecture of MaskMA naturally generalizes to tasks with variable numbers of agents. However, as most multi-agent tasks involve actions that represent interaction among units, *e.g.*, the healing and attacking in SMAC. Therefore each action length also grows up with the unit number.

We propose Generalizable Action Representation (GAR) to enable the capacity of MaskMA in dealing with the action space that varies according to unit number. Given an action $a_i^t$ that involves interaction between two units $u_i$ and $u_j$, we define $u_i$ as the executor of $a_i^t$ and $u_j$ the receiver. The embedding $E\left(a_i^t\right)$ of $a_i^t$ is defined as $E\left(a_i^t\right) = h_i^t \oplus h_j^t$, where $h_i^t$ and $h_j^t$ are the output embedding of $u_i$ and $u_j$ from the encoder, and $\oplus$ denotes the concatenation operation. With the $E\left(a_i^t\right)$ defined above, we generate the logits of interactive action by $FC\left(E\left(a_i^t\right)\right)$, with $FC$ denoting a fully-connected layer, and use $FC\left(h_i^t\right)$ for actions that do not involve interaction. These logits are then combined and fed into a softmax function to obtain the final action.

## 4 EXPERIMENTS

In this section, we design experiments to evaluate the following features of MaskMA. (1) Zero-shot and convergence of MaskMA. We conduct experiments on SMAC using only 11 maps for training and up to 60 maps for testing, assessing the model's ability to generalize to unseen scenarios. In SMAC tasks, agents must adeptly execute a set of skills such as alternating fire, kiting, focus fire, and positioning to secure victory. These attributes make zero-shot transfer profoundly challenging. (2) Effectiveness of mask-based collaborative learning strategy and generalizable action representation for different multi-agent tasks. We conduct ablation studies to find out how the sequence modeling forms of MaskMA affect performance and how the training strategy and generalizable action representation boost the generalization of MaskMA. (3) Generalization of MaskMA to downstream tasks. We

Table 3: **Varied Policies Collaboration on 8m_vs_9m.** Cooperating with a different performance player who achieves a 41% win rate, MaskMA demonstrates excellent collaborative performance in diverse scenarios with varying numbers of agents with varied performance.

| # Agents with varied performance | 0 | 2 | 4 | 6 | 8 |
|---|---|---|---|---|---|
| Win rate | 86.72±4.06 | 89.84±2.59 | 79.69±5.18 | 62.50±7.33 | 41.41±6.00 |

Table 4: **Teammate Malfunction on 8m_vs_9m.** "Marine Malfunction Time" indicates the time of a marine malfunction during an episode. For instance, a value of 0.2 means that one marine begins to exhibit a stationary behavior at 1/5th of the episode. Entry 1.0 signifies the original 8m_vs_9m configuration without any marine malfunctions.

| Marine Malfunction Time | 0.2 | 0.4 | 0.6 | 0.8 | 1.0 |
|---|---|---|---|---|---|
| Win Rate | 1.56±1.56 | 37.5±6.99 | 71.09±6.77 | 86.72±2.59 | 86.72±4.06 |

evaluate the model's performance on various downstream tasks, such as varied policies collaboration, teammate malfunction, and ad hoc team play. This helps us understand how the learned skills and strategies can be effectively adapted to different situations.

**Setup**. In SMAC (Samvelyan et al., 2019), players control ally units in StarCraft using cooperative micro-tricks to defeat enemy units with built-in rules. Our approach differs from existing methods that only consider grouped scenarios, such as Easy, Hard, and Super-Hard maps. Instead, we extend the multi-agent decision-making tasks by combining different units with varying numbers. We include three races: Protoss (colossus, zealot, stalker), Terran (marauder, marine, and medivac), and Zerg (baneling, zergling, and hydralisk). Note that since StarCraft II does not allow units from different races to be on the same team, we have designed our experiments within this constraint. Firstly, we collect expert trajectories as offline datasets from the 11 training maps by utilizing the expert policies trained with a strong RL method named ACE (Li et al., 2022) This yields 11 offline datasets, most of which contain 10k episodes with an average return exceeding 18. Then, we employ different methods to pretrain on the offline dataset and evaluate their zero-shot capabilities on 60 generated test maps. As shown in Table 1, we run 32 test episodes to obtain the win rate and report the average win rate as well as the standard deviation across 4 seeds. In the results we present, 'CE' stands for centralized execution, and 'DE' denotes decentralized execution. In cases where no specific notation is provided, the results are based on DE. We take the MADT method as our baseline for comparison which utilizes a causal transformer to consider the history of local observation and action for an agent.

### 4.1 PERFORMANCE ON PRETRAINING DATASETS

We assess MaskMA and baselines on offline datasets including 11 training maps. As shown in Table 1, MaskMA achieves a 90% average win rate in 11 maps both for CE and DE, while MADT only achieves a 51.78% win rate for DE and struggles in more challenging maps, even reaching a 14% win rate. One key observation from the results is that MaskMA consistently performs well in both centralized training centralized execution (CTCE) and centralized training decentralized execution (CTDE) settings, highlighting its flexibility and adaptability in various execution paradigms.

Figure 3a represents the testing curve of MaskMA and the baseline in 11 training maps. MaskMA significantly outperforms the baseline with lower variance and achieves more than 80% win rate in most maps within 0.5M training steps, showing the robustness and efficiency of MaskMA. While the mask-based collaborative learning strategy introduces a level of complexity that can cause a performance degradation compared to MaskMA without masking during the pretraining phase, it effectively forces MaskMA to adjust to varying ranges of observation, including both global and partial observations and learn robust representations that are beneficial for generalization.

### 4.2 MASKMA AS EXCELLENT ZERO-SHOT LEARNERS

We present the results of our MaskMA and the baseline on zero-shot learning tasks in multi-agent scenarios. Specifically, we evaluate different methods by the win rate on the 60 unseen test maps.

Table 5: **Ad hoc Team Play on 7m_vs_9m.** "Marine Inclusion Time" indicates the time of adding an additional marine during an episode. For example, a value of 0.2 represents adding one marine at 1/5th of the episode. Entry 1.0 signifies the original 7m_vs_9m setup without any additional marine.

| Marine Inclusion Time | 0.2 | 0.4 | 0.6 | 0.8 | 1.0 |
|---|---|---|---|---|---|
| Win Rate | 80.47±7.12 | 78.12±2.21 | 50.00±8.84 | 10.94±6.81 | 0±0 |

Table 6: **Ablation over mask-based collaborative learning strategy (MCLS) and generalizable action representation (GAR).** Baseline utilizes a transformer architecture. Each row adds a new component to the baseline, showcasing how each modification affects the overall performance.

| Setting | CE | DE |
|---|---|---|
| Transformer | 44.67±3.35 | 8.03 ±1.44 |
| + MCLS | 39.49±3.05 | 39.91±3.97 |
| + GAR | 91.26±4.21 | 41.55±4.38 |
| MaskMA (full model) | 90.91±3.56 | 89.35±3.92 |

Table 2 shows that MaskMA outperforms the baseline method in zero-shot scenarios by a large margin, successfully transferring knowledge to new tasks without requiring any additional fine-tuning. Specifically, MaskMA achieves a 79.71% win rate for CE and a 77.75% win rate for DE, while MADT only achieves a 43.72% win rate. These results indicate that MaskMA's mask-based collaborative learning strategy and generalizable action representation effectively address the challenges of partial observability and varying agent numbers and action spaces in multi-agent environments.

Furthermore, we can observe that MaskMA consistently performs well across varying levels of complexity, as demonstrated by the win rates in different entity groups. In contrast, MADT achieves limited performance with high variance across different entity groups. This highlights the ability of MaskMA to generalize and adapt to diverse scenarios, which is a key feature of a robust multi-agent decision, making it a versatile and reliable choice for multi-agent tasks.

## 4.3 Performance on Downstream Tasks

In this section, we provide various downstream tasks to further evaluate the strong generalization of MaskMA, including varied policies collaboration, teammate malfunction, and ad hoc team play.

**Varied Policies Collaboration**. In this task, partial agents are controlled by the best policy, and other agents are controlled by other policies with varied performance, as it requires generalized policies to coordinate with different operations at various levels. We conducted simulations using a model with average performance (win rate of 41%) to represent a player with a different policy in the 8m_vs_9m map, where our team controlled 8 marines to defeat 9 enemy marines. As shown in Table 3, MaskMA exhibits seamless collaboration with other agents under different scenarios where varying numbers of agents have different operations and performance. MaskMA dynamically adapts to the strategies of other players and effectively coordinates actions. Furthermore, when the number of agents with different performance is 8, MaskMA itself does not control any agents. Therefore, the win rate in this case can represent the win rate of the players controlled by different policies and humans.

**Teammate Malfunction**. In this task, teammates may get out of order or die due to external factors during inference. MaskMA is designed to handle such situations gracefully by redistributing tasks among the remaining agents and maintaining overall performance. As shown in Table 4, MaskMA exhibits robustness and adaptability in the face of unexpected teammate malfunction.

**Ad hoc Team Play**. In this task, agents need to quickly form a team with new agents during the execution of the task. The challenge lies in the ability of the model to incorporate new agents into the team and allow them to contribute effectively without disrupting the ongoing strategy. As shown in Table 5, MaskMA demonstrates excellent performance in ad hoc team play scenarios, adjusting its strategy to accommodate new agents and ensuring a cohesive team performance.

Table 7: **Mask type ablation.** We compare various mask types for pretraining with fixed ratios from 0 to 0.8 and random ratios. Env represents using the local visibility of the agent in the environment.

| Mask Type | 0 | 0.2 | 0.5 | 0.8 | Env | Random (0, 1) |
|---|---|---|---|---|---|---|
| CE | 91.26±4.21 | 89.70±3.81 | 88.21±3.78 | 82.81±4.83 | 55.97±4.67 | 90.91±3.56 |
| DE | 41.55±4.38 | 58.03±5.70 | 71.52±4.23 | 82.03±5.01 | 83.59±8.08 | 89.35±3.92 |

Overall, the results in this section demonstrate the versatility and generalization capabilities of MaskMA across various downstream tasks. These findings highlight the potential of MaskMA to advance the field of multi-agent and its applicability in real-world scenarios.

## 4.4 ABLATION STUDY

We perform ablation studies to access the contribution of each individual component: mask-based collaborative learning strategy and generalizable action representation. Our results are reported in Table 6 where we compare the performance of removing each component from MaskMA along with our modifications to the architecture. Furthermore, we conduct ablation studies to understand the influence of hyperparameters including timestep length and sight mask ratio.

**Generalizable Action Representation.** We ablate the generalizable action representation by comparing our proposed action space to an alternative action space, which aligns the maximum action length with a specific action mask for each downstream task. As shown in Table 6, removing the generalizable action space leads to significant performance degradation (row 4th and row 2nd), emphasizing its importance in improving the model's generalization capabilities.

**Mask-based Collaborative Learning Strategy.** Table 6 (row 4th and row 3rd) shows that the model without masked training struggles to generalize to new settings, exhibiting significant performance degradation. The mask-based collaborative learning strategy employed in MaskMA, while posing a challenging pretraining task, helps the model learn permanent representations that are useful for generalization. This is evident from the performance improvement in the CE setting, where MaskMA demonstrates a better capacity to adapt to local observation situations compared to the one without the mask-based collaborative learning strategy. Intuitively, the random mask ratio is consistent with the inference process, where the number of enemies and allies gradually increases in an agent's local observation due to cooperative micro-operations, such as positioning, kiting, and focusing fire.

It is important to note that our "Transformer" column in Table 6 essentially represents behavior cloning and our method outperforms behavior cloning by a significant margin. Furthermore, we provide mask ratio analysis as shown in Table 7. The results show that as the masking ratio increases, the performance of the model improves for decentralized execution (DE) while decreasing for centralized execution (CE). This suggests that an appropriate masking ratio helps strike a balance between learning useful representations and maintaining adaptability to dynamic scenarios in the agent's local observation. In conclusion, a random ratio mask is a simple yet effective way, considering the CE and DE, to absorb the advantages of various fixed ratio masks and env masks. This approach allows MaskMA to demonstrate strong performance in both centralized and decentralized settings while maintaining the adaptability and generalization necessary for complex multi-agent tasks.

**Timestep Length.** To assess the importance of access to previous states, we ablate on the timestep length $K$. As shown in Figure 3b, MaskMA performance is better when using a longer timestep length. One hypothesis is that the POMDP property of the SMAC environment necessitates that policies in SMAC take into account sufficient historical information in order to make informed decisions. Considering the balance between performance and efficiency, we use $K=10$ in other experiments. This choice allows MaskMA to leverage enough historical information to make well-informed decisions while maintaining a reasonable level of computational complexity.

**Zero-Shot Capability with Pretraining Map Numbers** Figure 3c demonstrates the relationship between zero-shot capability and the number of pretrained maps in MaskMA. As the number of training maps increases, the win rate also improves, indicating that the model is better equipped

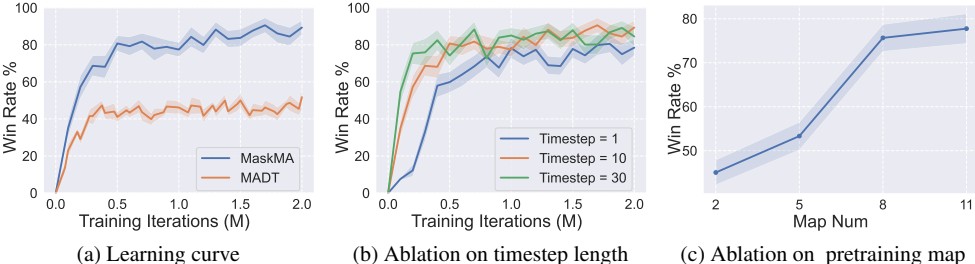

(a) Learning curve      (b) Ablation on timestep length      (c) Ablation on pretraining map

Figure 3: **(a) Learning curve.** MaskMA consistently outperforms MADT on average win rate in 11 training maps. **(b) Ablation on timestep length.** MaskMA performs better when using a longer timestep length. **(c) Ablation on pretraining map numbers.** With the increasing number of training maps, especially from 5 to 8, the model's performance on various unseen maps also improves, indicating better generalization to new tasks.

to tackle new situations. A marked uptick in win rate is observed when the map count rises from 5 to 8, underlining the value of training the model across varied settings. This trend in MaskMA offers exciting prospects for multi-agent decision-making. It implies that by augmenting the count of training maps or integrating richer, more intricate training scenarios, the model can bolster its adaptability and generalization skills.

**Training Cost and Parameter Numbers** MaskMA processes the inputs of all agents concurrently, achieving a notable degree of parallelism superior to MADT, which transforms multi-agent pretraining data into single-agent data. Consequently, MaskMA is considerably more time-efficient than MADT when trained over identical epochs. Specifically, MaskMA completes pretraining on 11 maps in 31 hours, whereas MADT requires 70 hours. For an equitable comparison, both MaskMA and MADT employ transformers of the same architecture. The sole distinction is in the final fully connected (FC) layer responsible for action output, making the parameter count for both models nearly identical.

## 5 LIMITATIONS AND FUTURE WORK

**Comparison to More Specialized Models** In our study, we focused on utilizing sequence modeling and masking strategies for Multi-Agent decision-making. Although we achieved promising results, comparing MaskMA with specialized models designed for specific tasks or environments could offer deeper insights. In the future, we aim to conduct a comprehensive evaluation of MaskMA against these specialized models to better understand the strengths and weaknesses of MaskMA.

**More Data with Different Quality** Our current evaluation was based on a limited dataset, which may not fully represent the diverse range of possible agent interactions and environmental conditions. We plan to explore the impact of different data qualities on the performance of our method. By including datasets with varying levels of noise, complexity, and agent behavior, we aim to gain a better understanding of our model's robustness and sensitivity to data quality. This will help us further refine MaskMA and enhance its performance in real-world scenarios with diverse data sources.

## 6 CONCLUSION

In this paper, we have addressed the challenges of zero-shot generalization and adaptability in multi-agent decision-making. To tackle these challenges, we introduced MaskMA, a masked pretraining framework for multi-agent decision-making that employs a transformer architecture, mask-based collaborative learning strategy, and generalizable action representation. Our proposed framework enables the model to learn effective representations and strategies by capturing the underlying correlations among agents and their actions while maintaining adaptability to dynamic scenarios. Extensive experiments on SMAC demonstrate the effectiveness of MaskMA in terms of zero-shot performance, generalization, and adaptability to various downstream tasks, such as varied policies collaboration, teammate malfunction, and ad hoc team play. Our findings encourage further exploration of more sophisticated masking strategies and efficient pretraining techniques for multi-agent decision-making.

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

In this Supplementary Material, we provide more elaboration on the implementation details and experiment results. Specifically, we present the implementation details of the model training in Section A and additional results and visualization in Section B.

# A  ADDITIONAL IMPLEMENTATION DETAILS

In this section, we provide a detailed description of the required environment, hyperparameters, and the specific composition of the entity's state for the SMAC. We will release our code, dataset, and pretrained model after this paper is accepted.

## A.1  ENVIRONMENT.

We use the following software versions:

- CentOS 7.9
- Python 3.8.5
- Pytorch 2.0.0
- StarCraft II 4.10

We conduct all experiments with a single A100 GPU.

## A.2  HYPERPARAMETERS

As shown in Table 8, our experiments of MaskMA and baseline MADT utilize the same hyperparameters.

Table 8: Hyperparameters of MaskMA and baseline MADT. It should be noted that both models utilize the exact same set of hyperparameters.

|  | Hyperparameter | Value |
|---|---|---|
| Training | Optimizer | RMSProp |
|  | Learning rate | 1e-4 |
|  | Batch size | 256 |
|  | Weight decay | 1e-5 |
| Architecture | Number of blocks | 6 |
|  | Hidden dim | 128 |
|  | Number of heads | 8 |
|  | Timestep length | 10 |

## A.3  STATE OF UNITS

In the original StarCraft II Multi-Agent Challenge (SMAC) setting, the length of the observation feature fluctuates in accordance with the number of agents. To enhance generalization, MaskMA directly utilizes each unit's state as input to the transformer architecture. As depicted in Table 9, within the SMAC context, each unit's state comprises 42 elements, constructed from nine distinct sections. Specifically, the unit type section, with a length of 10, represents the nine unit types along with an additional reserved type.

# B  ADDITIONAL RESULTS AND ANALYSIS

## B.1  WIN RATE OF ALL MAPS

As shown in Table 10, we present the win rate of MaskMA and MADT on 11 training maps and 60 testing maps.

Table 9: The composition of the state.

| State name | dim |
|---|---|
| ally or enemy | 1 |
| unit type | 10 |
| pos.x and y | 2 |
| health | 1 |
| shield | 2 |
| cooldown | 1 |
| last action | 7 |
| path | 9 |
| height | 9 |

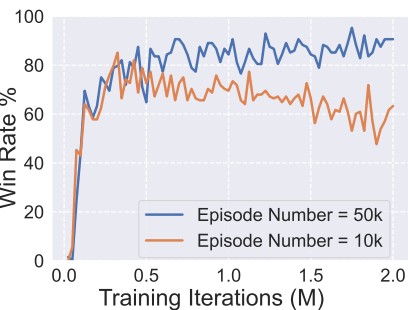

Figure 4: Performance comparison across different sizes of the pretraining dataset on the 3s_vs_5z Map.

## B.2 COMPARITION OF PRETRAINING DATASET SIZE

The scale of the pretraining dataset significantly impacts the eventual performance. In the multi-agent StarCraft II environment, SMAC, we investigated the optimal size for the pretraining dataset. We take the 3s_vs_5z map as an example and solely use the pretraining dataset of this map to train MaskMA, and then test it on the same map. As illustrated in Figure 4, a dataset encompassing 1k episodes was found insufficient, leading to a progressive decline in win rates. In contrast, a dataset comprising 50k episodes demonstrated exceptional performance. Specifically, for the 3s_vs_5z and MMM2 maps, a pretraining dataset containing 50k episodes proved appropriate. For the remaining nine maps, a dataset consisting of 10k episodes was found to be suitable.

## B.3 VISUALIZATION

In this section, we provide visualizations of MaskMA's behavior on three downstream tasks: varied policies collaboration, teammate malfunction, and ad hoc team play. Figure 5, 6, 7 evaluate the strong generalization of MaskMA. Additionally, we offer replay videos for a more comprehensive understanding of MaskMA's behavior and strategies.

Table 10: Win rate of MaskMA and MADT on 11 training maps and 60 testing maps.

| Map | Ours | | MADT |
|---|---|---|---|
| | CE | DE | DE |
| 3s_vs_5z | 85.94 ±3.49 | 82.81 ±7.81 | 73.44 ±3.49 |
| 3s5z | 98.44 ±1.56 | 99.22 ±1.35 | 15.62 ±6.99 |
| 1c3s5z | 94.53 ±4.06 | 95.31 ±1.56 | 54.69 ±8.41 |
| 3s5z_vs_3s6z | 85.94 ±6.44 | 85.16 ±5.58 | 14.84 ±9.97 |
| 5m_vs_6m | 86.72 ±1.35 | 84.38 ±4.94 | 85.94 ±5.18 |
| 8m_vs_9m | 88.28 ±6.00 | 86.72 ±4.06 | 87.50 ±2.21 |
| MMM2 | 92.97 ±2.59 | 86.72 ±4.62 | 62.50 ±11.69 |
| 2c_vs_64zg | 99.22 ±1.35 | 92.97 ±2.59 | 34.38 ±9.11 |
| corridor | 96.88 ±3.83 | 94.53 ±2.59 | 21.88 ±11.48 |
| 6h_vs_8z | 75.00 ±5.85 | 76.56 ±6.44 | 27.34 ±6.77 |
| bane_vs_bane | 96.09 ±2.59 | 98.44 ±1.56 | 91.41 ±4.62 |
| 2s_vs_1z | 100.00±0.00 | 100.00±0.00 | 70.31 ±10.00 |
| 3z_vs_3s | 71.88 ±3.83 | 53.91 ±9.47 | 0.00 ±0.00 |
| 1c2s | 44.53 ±5.58 | 41.41 ±6.39 | 2.34 ±2.59 |
| 4z_vs_3s | 99.22 ±1.35 | 97.66 ±1.35 | 4.69 ±2.71 |
| 4m | 99.22 ±1.35 | 100.00±0.00 | 28.91 ±6.00 |
| 1c3s | 54.69 ±3.49 | 48.44 ±4.69 | 17.19 ±4.69 |
| 5m_vs_4m | 100.00±0.00 | 100.00±0.00 | 100.00±0.00 |
| 5m | 100.00±0.00 | 100.00±0.00 | 98.44 ±1.56 |
| 2s3z | 10.94 ±5.18 | 8.59 ±2.59 | 0.00 ±0.00 |
| 1s4z_vs_1ma4m | 91.41 ±4.62 | 88.59 ±3.41 | 35.16 ±3.41 |
| 3s2z_vs_2s3z | 9.38 ±5.85 | 11.72 ±2.59 | 0.00 ±0.00 |
| 2s3z_vs_1ma2m2me | 92.97 ±4.06 | 82.03 ±14.04 | 35.16 ±9.73 |
| 2s3z_vs_1ma3m1me | 90.62 ±3.12 | 93.75 ±3.83 | 37.50 ±7.97 |
| 1c3s1z_vs_1ma4m | 100.00±0.00 | 99.22 ±1.35 | 98.44 ±1.56 |
| 1s4z_vs_2ma3m | 83.59 ±7.12 | 77.66 ±3.41 | 73.44 ±6.44 |
| 1c3s1z_vs_2s3z | 100.00±0.00 | 100.00±0.00 | 78.12 ±5.85 |
| 2s3z_vs_1h4zg | 96.88 ±3.12 | 96.09 ±3.41 | 71.88 ±7.16 |
| 3s2z_vs_2h3zg | 95.31 ±5.18 | 93.75 ±4.42 | 75.00 ±3.83 |
| 1c4z_vs_2s3z | 100.00±0.00 | 100.00±0.00 | 71.88 ±6.63 |
| 2ma3m_vs_2ma2m1me | 46.09 ±12.57 | 39.06 ±9.24 | 0.00 ±0.00 |
| 2s3z_vs_1b1h3zg | 100.00±0.00 | 96.88 ±2.21 | 67.19 ±4.06 |
| 1s4z_vs_5z | 3.12 ±2.21 | 5.47 ±4.06 | 0.00 ±0.00 |
| 1c1s3z_vs_1c4z | 64.06 ±7.16 | 76.56 ±5.63 | 35.94 ±6.44 |
| 6m | 97.66 ±2.59 | 99.22 ±1.35 | 47.66 ±5.12 |
| 1s3z_vs_5b5zg | 93.75 ±3.83 | 87.50 ±2.21 | 58.59 ±1.35 |
| 8z_vs_6h | 93.75 ±2.21 | 98.44 ±1.56 | 38.28 ±4.06 |
| 2s5z | 17.97 ±4.06 | 7.03 ±3.41 | 0.78 ±1.35 |
| 1c3s1z_vs_1b1h8zg | 100.00±0.00 | 100.00±0.00 | 57.81 ±0.00 |
| 1c3s1z_vs_1h9zg | 100.00±0.00 | 100.00±0.00 | 60.94 ±0.00 |
| 2c1s2z_vs_1h9zg | 100.00±0.00 | 100.00±0.00 | 49.22 ±0.00 |
| 3ma6m1me_vs_3h2zg | 100.00±0.00 | 100.00±0.00 | 23.44 ±0.00 |
| 3s2z_vs_1b1h8zg | 97.66 ±1.35 | 85.94 ±4.69 | 52.34 ±2.59 |
| 5ma4m1me_vs_1b3h1zg | 100.00±0.00 | 100.00±0.00 | 98.44 ±2.71 |
| 4ma5m1me_vs_5zg | 100.00±0.00 | 100.00±0.00 | 85.94 ±8.12 |
| 1s5z_vs_4ma6m | 39.06 ±10.00 | 16.41 ±3.41 | 4.69 ±4.69 |
| 4ma7m_vs_2s3z | 98.44 ±1.56 | 100.00±0.00 | 35.16 ±13.07 |
| 2ma9m_vs_3s2z | 89.06 ±2.71 | 80.47 ±6.00 | 51.56 ±9.50 |
| 5s2z_vs_2ma8m | 48.44 ±9.24 | 29.69 ±11.16 | 8.59 ±4.06 |
| 2s5z_vs_9m1me | 99.22 ±1.35 | 99.22 ±1.35 | 46.09 ±4.94 |
| 1c3s_vs_3h10zg | 85.94 ±5.63 | 89.84 ±5.58 | 43.75 ±4.94 |
| 3s7z_vs_5ma2m3me | 99.22 ±1.35 | 96.88 ±3.83 | 60.94 ±2.59 |
| 1c3s6z | 58.59 ±7.12 | 61.72 ±7.77 | 14.84 ±6.77 |
| 2c2s6z_vs_1c3s6z | 100.00±0.00 | 100.00±0.00 | 97.66 ±1.35 |
| 1c2s7z | 25.00 ±9.63 | 25.78 ±5.12 | 7.81 ±3.49 |
| 1c4s5z_vs_5ma3m2me | 100.00±0.00 | 83.59 ±8.08 | 84.38 ±0.00 |
| 10s1z_vs_1b2h7zg | 96.09 ±4.06 | 96.88 ±2.21 | 68.75 ±1.56 |
| 10m_vs_11m | 79.69 ±6.44 | 78.91 ±7.45 | 0.00 ±0.00 |
| 1b10zg | 85.16 ±5.58 | 89.06 ±7.16 | 60.16 ±5.12 |
| 2b10zg | 92.19 ±4.69 | 98.44 ±1.56 | 30.47 ±5.58 |
| 1c_vs_32zg | 59.38 ±9.38 | 55.47 ±7.77 | 1.56 ±2.71 |
| 32zg_vs_1c | 92.97 ±3.41 | 94.53 ±3.41 | 53.91 ±3.12 |
| 1b20zg | 74.22 ±10.91 | 82.81 ±2.71 | 66.41 ±7.45 |
| 2b20zg | 100.00±0.00 | 97.66 ±1.35 | 71.09 ±4.94 |
| 3b20zg | 96.09 ±3.41 | 88.28 ±4.62 | 85.94 ±4.69 |
| 16z_vs_6h24zg | 97.66 ±2.59 | 97.66 ±2.59 | 71.09 ±4.06 |
| 5b20zg | 99.22 ±1.35 | 100.00±0.00 | 32.03 ±2.59 |
| 64zg_vs_2c | 55.47 ±6.39 | 56.25 ±2.21 | 48.44 ±8.41 |
| 1c8z_vs_64zg | 89.06 ±7.16 | 85.94 ±5.18 | 2.34 ±2.59 |
| 1c8z_vs_2b64zg | 61.72 ±7.77 | 65.62 ±5.85 | 0.00 ±0.00 |
| 1c8z_vs_5b64zg | 6.25 ±2.21 | 4.69 ±3.49 | 0.78 ±1.35 |

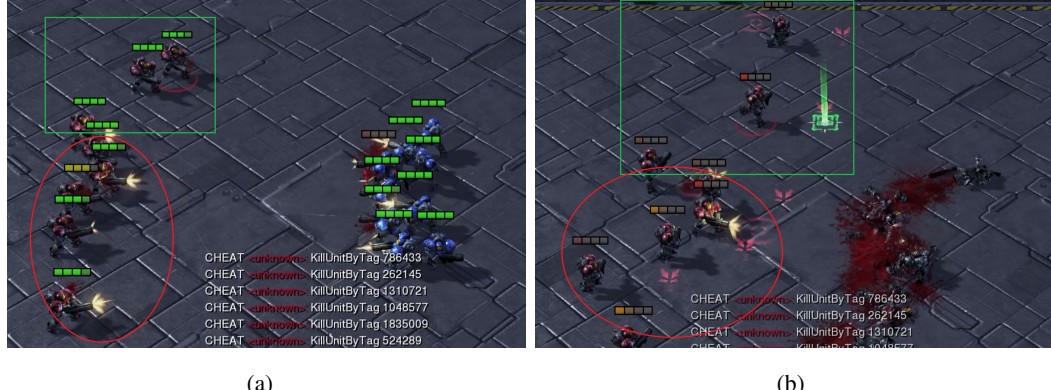

(a)                      (b)

Figure 5: Varied Policies Collaboration on 8m_vs_9m with 4 Agents with different policies. The agents within the green box are controlled by other policies (replaced with a network trained with a 41% win rate), while the agents within the red circle are controlled by MaskMA. **(a)**: Initial distribution of agents' positions. **(b)**: MaskMA dynamically adapts to the strategies of players by different policies and effectively coordinates actions, resulting in a victorious outcome.

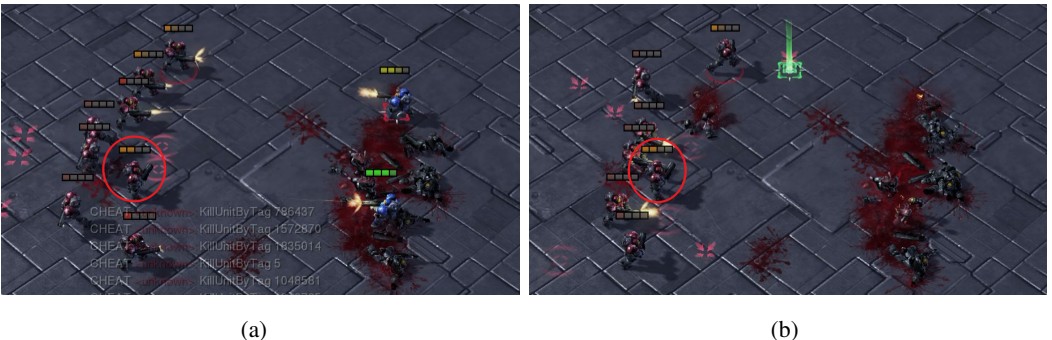

(a)                      (b)

Figure 6: Teammate Malfunction on 8m_vs_9m with Marine Malfunction Time = 0.6. The agent within the red circle suddenly malfunctions in the middle of the episode, remaining stationary and taking no actions. **(a)**: The agent within the red circle starts malfunctioning. **(b)**: Despite the malfunctioning teammate, other agents continue to collaborate effectively and eventually succeed in eliminating the enemy.

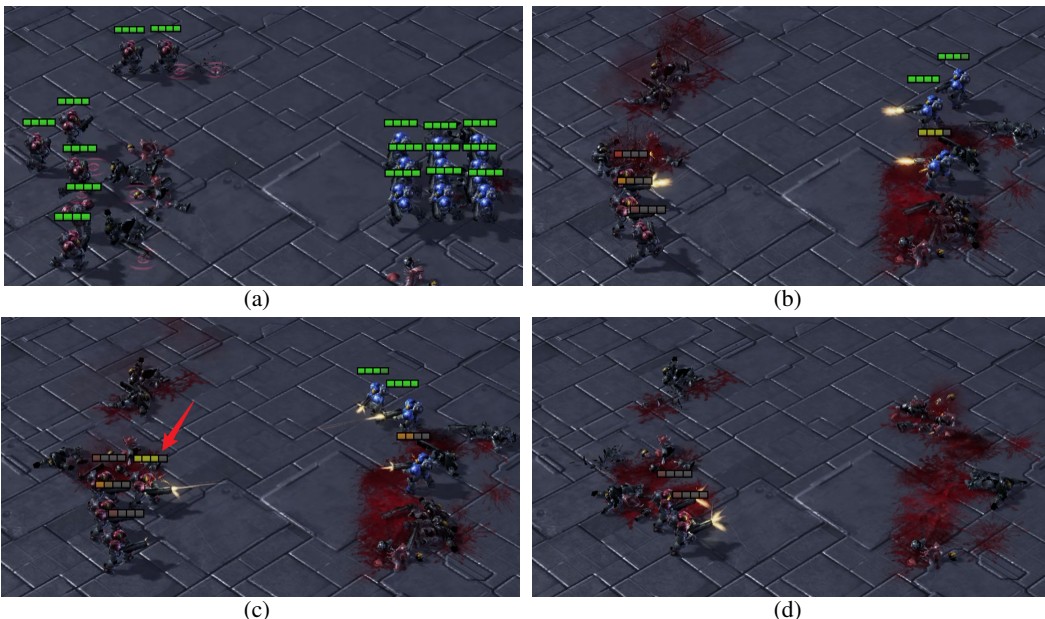

(a)        (b)

(c)        (d)

Figure 7: Ad hoc Team Play on 7m_vs_9m with Marine Inclusion Time = 0.8. This experiment demonstrates that when new Marines are added near the end of an episode, MaskMA still can quickly incorporate them into the team and enable them to contribute effectively. **(a)**: Initial distribution of agents' positions. **(b)**: Prior to the addition of the new Marine, our team is left with only three severely wounded agents, on the brink of defeat. **(c)**: The new agent (indicated by the red arrow) joins our team and immediately engages the enemy. **(d)**: With the assistance of the newly added agent, our team successfully defeats the enemy.

