# OpenReview forum: "Masked Pretraining for Multi-Agent Decision Making"
_ICLR.cc/2024/Conference — ICLR 2024 Conference Withdrawn Submission_

### Official Review · Reviewer_cZoR · 2023-10-28

**Soundness:** 2 fair
**Presentation:** 2 fair
**Contribution:** 2 fair
**Rating:** 5
**Confidence:** 3

**Summary:**

This paper extends MADT by randomly masking agents to address the gap between centralized training and decentralized execution, and by introducing actions performed by one unit u_i to another unit u_j with FC(E(u_i, u_j)), to achieve scalable action representations with the varying number of agents.

**Strengths:**

This paper is the first multi-agent pretraining model for decision-making with strong zero-shot performance.

**Weaknesses:**

I am not an expert in MARL. Due to a lack of background, I found the paper very unclear. I will list the questions in the next section.

**Questions:**

1. What is the architecture of MaskMA? Is it also based on a Decision Transformer? I didn't find value as input in the transformer architecture shown in Figure 2. How did you train MaskMA? Is it similar to supervised learning in a Decision transformer?
2. What is the input for each agent? I can see the description "includes unit type, position, health, shield, and so on". But do you have pixel input for each agent as shown in Figure 2? What encoder did you use for the pixel input?
3. The zero-shot performance seems to be very impressive. But why it can generalized across different environments? In recent pretraining works for embodied AI, their models can generalize because they have natural language instructions and they learn on multiple skills and objects, making them able to generalize to unseen combinations of seen skills and seen objects. I was wondering what the foundation models in MaskMA learned, and why it can generalize so well with zero-shot learning.
4. Do you have the map information? Is the map information introduced as context for different tasks? Does the map information really make a difference in agents' decisions? Or is the agent's decision mainly based on the information of other units? If so, I think a map is not that important, and the generalization ability can be explained by the masking mechanism to deal with varying numbers of different other agents/units.
5. The action representation is good for starcraft, where units interact with each other through actions, but may not be general to other MARL environments.
I will consider raising my score if all the questions are addressed.

---

### Official Review · Reviewer_Y6zM · 2023-10-29

**Soundness:** 3 good
**Presentation:** 2 fair
**Contribution:** 2 fair
**Rating:** 3
**Confidence:** 4

**Summary:**

This paper proposes a Masked pretraining framework to address the challenges of zero-shot capabilities and generalization in multi-agent scenarios. The authors introduce two scalable techniques: Mask-based Collaborative Learning Strategy (MCLS) and Generalizable Action Representation (GAR). MCLS incorporates random masking into the attention matrix of the transformer to reconcile the discrepancy between centralized pretraining and partial observations. GAR divides the action space into actions toward self-information and actions related to other entities, allowing MaskMA to handle tasks with varying agent numbers and action spaces.
The experiment results are impressive. A single model pretrained on 11 training maps achieves a 77.8% zero-shot win rate on 60 unseen test maps. The method also shows effective generalization in various downstream tasks.

**Strengths:**

**Method:** The proposed method MaskMA is simple yet effcient. The experimental results shows impressive zero-shot generation ability in 60 testing maps with one model pretrained on 11 maps.

**Data:** The authors provide a wide range of maps than original SAMC for testing.

**Problem Setting:** Except for zero-shot generation problem, this paper also provides some novel and interesting problem settings which includes varied policies collaboration, teammate sudden malfunction and sudden addition teammate.

**Weaknesses:**

**Novelty:** Although the proposed method MaskMA seem to be very effective in zero-shot generation for MA decision making, this method is lack of novelty. The main architecture is based on Transformer with two key components Mask-based Collaborative Learning Strategy (MCLS) and Generalizable Action Representation (GAR).  MCLS is essentially a random mask on the attention matrix which is more like a multi-task pretraining trick which can provide multiple battle scenarios during pretraining. The idea of GAR to group actions is common, and GAR also lacks certain theoretical explanations.

**Experiment:** Compared algos only contain MADT which is insufficient. Except for zero-shot generation experiment, the results in section *PERFORMANCE ON DOWNSTREAM TASKS* are lack of comparision with other alogs and settings. For example, in *Table 5*, including a Marine in the 1/5th of the episode results in 80% win rate, however, without comparision, it is difficult to tell whether the results are good enough.

**Writing:** Overall, the writing of this paper can convey clear ideas. But there are a few parts need to be improve. First, Figures 1 and 2 are not referenced in the paper, which is confusing. Second, matrix J2 in section 3.2 has no shape description. And also it will be better to explain which sepcific part being masked after the maths to help reader to understand. Third, Figure 1 doesn't show the training part, like interaction with envs and the gradients flow. Last but not least, figures and tables are poorly laid out, often far from the text that references them.

**Questions:**

1. Why GAR is such effective? The authors should give more detailed explanation.
2. How MaskMA performs well without other entitys information in the states? The original SMAC obs or states all contain allys or enemys information while the state used in MaskMA only contains self information. Although the attention and GAR seem to help include information about others, there are still doubts about the performance.
3. Noticed there are several maps with extrame low win rate in Table 10, why MaskMA suffers low generation performance on these maps?

---

### Official Review · Reviewer_FVLE · 2023-10-31

**Soundness:** 3 good
**Presentation:** 2 fair
**Contribution:** 2 fair
**Rating:** 3
**Confidence:** 4

**Summary:**

This paper introduces a masked pretraining framework, MaskMA, to address the cooperative multi-agent RL problems, featuring two key components: a mask-based collaborative learning strategy (MCLS) and a generalizable action representation (GAR). The authors trained the model on 11 SMAC maps and conducted zero-shot tests on 60 unseen maps. Compared to previous algorithms based on the Transformer architecture, MaskMA significantly improved the zero-shot performance.

**Strengths:**

The article is easy to follow and employs a straightforward method that yields good results. After training on a small number of maps in StarCraft, it can generalize in a zero-shot manner to 60 broader unseen maps, showing significant improvement over previous baseline algorithms. The experimental results are interesting and impressive.

**Weaknesses:**

The novelty of this paper is somewhat limited, as random masking has been used previously in Bert and some RL tasks [1]. The GAR component appears to address only the issue of variable action spaces and incorporates some domain knowledge from StarCraft for action grouping. The baseline comparisons in the article are also limited, considering only Transformer models in sequence modeling and not directly comparing with related works that employ attention mechanisms, although they are discussed in the related work section, such as Uni[Mask], UPDeT, and ASN, which could be applicable to this paper's setting.

Another major concern is that, despite the large number of test environments, the experimental domain is quite narrow, i.e., only on SMAC. It would be beneficial to conduct supplementary tests in commonly used environments like Google Research Football (GRF). The paper does not demonstrate the connection and differences between the selected tasks and the tasks for generalization tests, such as why specifically these 11 tasks were chosen and what characteristics the 60 test tasks have, or how they differ from the selected tasks. I think foundation models are an important research direction, and the paper's demonstration of zero-shot generalization is impressive in terms of the number of training and testing tasks. However, the number of unseen tasks is not the most crucial metric. A more careful analysis of how tasks are chosen and on which conditions zero-shot generalization ability can be achieved would make the paper more solid.

Additionally, the selection of downstream tasks appears somewhat arbitrary, also without clear reasoning for the choice of corresponding maps (e.g., 8m vs 9m, 7m vs 9m).



In terms of writing, the notation in the paper may incur ambiguity, particularly in Section 3.2, where some notation are not clearly-defined (as detailed in the question part). The organization of figures and tables is somewhat chaotic, with some tables located several pages away from the relevant text, requiring back-and-forth scrolling. Besides, it would be better if the authors could use bold or other emphasis to highlight results in the tables.



References:

1. *Masked Trajectory Models for Prediction, Representation, and Control.* ICML 2023

**Questions:**

1. Why were these 11 tasks chosen instead of others? What is the relationship between these tasks and the 60 unseen tasks tested?
2. Some works in the related work section, such as Uni[mask] and UPDeT, seem to be directly applicable to this paper's setting; why were they not included in the comparisons?
3. Why were these particular downstream tasks chosen to demonstrate the benefits of MaskMA, and why were only the maps 8m vs 9m and 7m vs 9m used instead of others?
4. I noticed that the number of expert trajectories used is 10k or 50k (averaging about 18k according to the authors), which seems several times more than the 2k trajectories used in ODIS. Is this amount excessive, or how should the number of trajectories be chosen?
5. Section 3 states "Such an independent learning pipeline leads to an increase in computational complexity of $O(N^3)$"; why is the complexity $O(N^3)$?
6. In Section 3.2, each $J_1$ is an $N \times N$ matrix; how is the diagonal matrix $m_1\$ constructed with the $\mathtt{Diag}$ operation, and do the subsequent disjunction and conjunction symbols correspond directly to the binary "or" and "and" operations for 0-1 operations? If so, it is suggested to explain more to avoid ambiguity.
7. In the MaskMA and the Transformer structure presented in this paper, are the observations and actions of teammates included as inputs? If so, how is the order of agents determined?
8. Why were 4 seeds used in the experiments? The common practice is to use 5 or 10 seeds in SMAC.
9. In the ablation study of the number of pretraining maps, which 2, 5, or 8 maps were used, and were the same 60 unseen tasks always the test set?
10. According to the experimental results in Table 6, +MCLS actually reduced performance in CE mode, while GAR showed significant improvement. Is there any in-depth analysis?

---

### Official Review · Reviewer_famv · 2023-11-01

**Soundness:** 3 good
**Presentation:** 2 fair
**Contribution:** 2 fair
**Rating:** 5
**Confidence:** 3

**Summary:**

This paper develops MaskMA to improve the zero-shot capability for MARL. In particular, MaskMA addresses the mismatch between centralized training and decentralized execution by developing MCLS and handles the varying numbers of agents and actions across downstream tasks by developing GAR. Through extensive experiments on the SMAC benchmark, MaskMA shows a better zero-shot performance than the MADT baseline.

**Strengths:**

1. This paper performs extensive experiments on SMAC with interesting downstream tasks (Section 4.3).
2. MaskMA is a scalable framework w.r.t. the number of agents as shown in Table 2.

**Weaknesses:**

1. The novelty of MCLS could be limited w.r.t. existing work: the idea of mask-based learning in Section 3.2 could be similar to mask-based transformer (Cheng et al., 2022; Nie et al., 2023), where they also apply masked training for computer vision tasks. Given these related methods (in addition to MaskDP and Uni[MASK] in Section 2), it is unclear how new the idea of MCLS (i.e., randomly making part of the units in $s^t$) is.
2. GAR requires prior knowledge about which actions affect the environment and which actions correspond to agent interactions, so GAR is not completely model-free.
3. Only MADT is compared as a baseline. It is also unclear whether MADT is a SOTA method given that the work was published in 2021. Adding more baselines (including UPDeT as mentioned in Section 2) could help justify whether MaskMA achieves SOTA zero-shot MARL performance. Also, having more than one benchmark domain (other than SMAC) would make the experimental result more strong.

References:
Bowen Cheng, Ishan Misra, Alexander G. Schwing, Alexander Kirillov, Rohit Girdhar. Masked-attention Mask Transformer for Universal Image Segmentation. CVPR, 2022.
Hongkai Zheng, Weili Nie, Arash Vahdat, Anima Anandkumar. Fast Training of Diffusion Models with Masked Transformers. 2023.

**Questions:**

I hope to ask the authors' responses to my concerns (please refer to the weaknesses section for details).

---

### Official Review · Reviewer_pTAk · 2023-11-02

**Soundness:** 3 good
**Presentation:** 3 good
**Contribution:** 3 good
**Rating:** 6
**Confidence:** 2

**Summary:**

This paper presents a multi-agent RL method that learns by masking parts of a global state instead of training separate policies that take input states, which also includes a generalizable action representation technique that separately represents actions that involve multi-entity interaction from actions that does not. Experiments are conducted mainly on SMAC environment where the proposed method is compared against the baseline MADT that transforms multi-agent data to single-agent data.

**Strengths:**

- I'm not an expert in this area, but the paper was very easy to read and gives sufficient information helpful for undertstanding the main idea.
- Masking the partial inputs of global states and utilizing the transformer's capability that can handle variable length inputs makes sense and the reported results are quite promising.
- Experiments are extensively conducted.

**Weaknesses:**

- I'm not an expert in this area so not sure on this point, but there is only one baseline and the reason of only reporting the performance of sole baseline is not clear from the paper. Even if it's not a fair comparison, it could be nice to include other baseline performances and clearly explaining the results can be helpful.
- In terms of presentation this paper could be improved by better locating the figures and tables. Current arrangement requires the readers to scroll up and down quite a lot, which harms the readability.
- There are several typos, like casual -> causal, missing . in ACE (Li et al., 2022) This, not consistent usage of italic in FC, inconsistent use of . in paragraph (e.g., Action Representation.).

**Questions:**

- Please address & answer the comments in Weaknesses.
- Is there a reason why there is no baseline in downstream task experiments?